# What caused ozone pollution during the 2022 Shanghai lockdown? Insights from ground and satellite observations

Yue Tan and Tao Wang*

Department of Civil and Environmental Engineering, The Hong Kong Polytechnic University, Hong Kong, 999077, China

*Correspondence to*: Tao Wang (tao.wang@polyu.edu.hk)

**Abstract.** Shanghai, one of China's most important economic centres, imposed a citywide lockdown in April and May 2022 to contain a resurgence in cases of coronavirus disease 2019. Compared with the 2020 lockdown, the 2022 lockdown occurred in a warm season and lasted much longer, thereby serving as a relevant real-world test of the response of ambient ozone ($O_3$) concentrations to emission reductions in a high-$O_3$ season. In this study, we analysed surface observations of $O_3$ and nitrogen dioxide ($NO_2$) concentrations and satellite-retrieved tropospheric $NO_2$ and formaldehyde (HCHO) column concentrations in the first 5 months of 2022 with comparisons to the year 2021. During the 2-month 2022 lockdown, the maximum daily 8-h average (MDA8) $O_3$ concentrations at one or more of the city's 19 sites exceeded China's air quality standard of 160 μg/m$^3$ 21 times, with the highest value being 200 μg/m$^3$. The city-average MDA8 $O_3$ concentration increased by 13% in April–May 2022 year-on-year, despite sharp declines in $NO_2$ surface and column concentrations (both by 49%) and a 19% decrease in the HCHO column concentration. These results show that the reductions in $O_3$ precursors and other pollutants during the 2022 lockdown did not prevent ground-level $O_3$ pollution. An analysis of meteorological data indicates that there were only small changes in the meteorological conditions and there was little transport of $O_3$ from the high-$O_3$ inland regions during the 2022 lockdown, neither of which can account for the increased and high concentrations of $O_3$ that were observed during this period. The mean HCHO/$NO_2$ ratio in April–May increased from 1.11 in 2021 to 1.68 in 2022, and the correlation between surface $O_3$ and $NO_2$ concentrations changed from negative in 2021 to positive in 2022. These results indicate that the high $O_3$ concentrations in 2022 were mainly due to large

reductions in the emissions of $NO_x$ and that the decrease in the concentrations of volatile organic compounds

(VOCs) could not overcome the NO titration effect. During the 2022 lockdown, Shanghai's urban centre

remained VOC-sensitive despite drastic reductions in road transportation (73%-85%) and industrial

emissions (~40%), whereas its semi-rural areas transitioned from VOC-limited to VOC–$NO_x$ co-limited

regimes. Our findings suggest that future emission reductions similar to those that occurred during the

lockdown, such as those that will result from electrifying transportation, will not be sufficient to eliminate

$O_3$ pollution in urban areas of Shanghai and possibly other VOC-limited metropolis, without the imposition

of additional VOC controls or more substantial decreases in $NO_x$ emissions.

**1 Introduction**

Shanghai is a megacity with over 25 million residents and a land area of 6,341 $km^2$. It serves as a financial,

transport and logistics centre of mainland China. From 27 March to 1 June 2022, the city imposed a strict 2-

month lockdown (LCD) to curb increases in the number of cases of coronavirus disease 2019 (COVID-19).

Official statistics indicate that both economic activities and human livelihood were severely disrupted by the

2022 Shanghai LCD (hereinafter '2022 LCD'). For example, in April, year-on-year road passenger traffic

turnover decreased by 85.5% and road cargo turnover decreased by 73.1% (Shanghai Bureau of Statistics,

2022); total industrial output decreased by 61.6% and power generation decreased by 41.7% (National Bureau

of Statistics, 2022); and port cargo throughput at the Port of Shanghai, the world's largest seaport, decreased

by 36.5% (Ministry of Transport of the People's Republic of China, 2022). As Shanghai is the most important

transportation and logistics hub of the manufacturing-intensive Yangtze River Delta (YRD) region, the ripple

effects of the LCD in Shanghai have disrupted supply chains of its surrounding provinces, other regions of

China, and the world (Cao et al., 2022; Hale et al., 2022; He, 2022).

Such a reduction in human activity can be expected to drastically decrease emissions of air pollutants, as has

been borne out by numerous studies of the 2020 LCD (Wang et al., 2022; Huang et al., 2021; Doumbia et al.,

2021). Compared with the 2020 LCD in China, which was imposed in late January–early February, the 2022

LCD in Shanghai was imposed in the high-$O_3$ season and lasted for longer, during which time the $O_3$ concentrations exceeding the air quality standard were frequently observed. In the present study, we aimed to understand how meteorology and non-linear chemistry influenced ground-level $O_3$ concentrations during the 2022 LCD and whether the huge decrease in $NO_x$ emission could push a typical VOC-limited megacity to a $NO_x$-limited regime. We first analysed surface observations of $O_3$ and $NO_2$ concentrations and satellite-measured $NO_2$ and HCHO column concentrations to assess changes in the concentrations of these species in Shanghai and its surrounding areas during the 2022 LCD. We then examined the roles of meteorological and chemical conditions in determining $O_3$ concentrations during the 2022 LCD by analysing small- and large-scale meteorological data and chemical indicators of $O_3$-formation regimes. We discuss the implications of our findings for future strategies aimed at reducing $O_3$ concentrations, which is currently the dominant air pollutant in China in warm seasons.

**2 Data and methodology**

**2.1 Surface measurement data**

We obtained hourly concentrations of ground-level $O_3$, $NO_2$ and particulate matter of 2.5 μm and smaller ($PM_{2.5}$) recorded at ~1,700 stations in China during 2019–2022 from the China National Environmental Monitoring Centre (http://106.37.208.233:20035/). The ambient concentrations of $O_3$, $NO_2$ and $PM_{2.5}$ are measured by an automated monitoring system at each site and reported to the China National Environmental Monitoring Centre and published online after validation (Wang et al., 2014). $O_3$, $NO_2$, and $PM_{2.5}$ are measured with UV photometry, chemiluminescence, and micro-oscillating balance or β absorption, respectively, with a detection limit of 2 ppb, 2 ppb and 2 μg/m$^3$, respectively (https://www.mee.gov.cn). The national network has 10 stations in Shanghai in 2019-2020, most of which are situated within the city centre. Since 2021, the number of stations has increased to 20 with most newly added sites locating outside the city centre (refer to Fig. 1 and Fig. S1 for their locations). (Note there were no data from one site (Dianshanhu) in 2021-2022, thus the data from 19 stations are available for years of 2021 and 2022). The maximum daily 8-h average

(MDA8) $O_3$ concentration was calculated for each site as the highest value of the 24 8-h moving average $O_3$ concentrations for a given day. The daily average concentrations of the other pollutants ($NO_2$ and $PM_{2.5}$) were also calculated from the hourly data. The 14-day moving averages of the surface pollutant concentrations were calculated, as these had fewer fluctuations than daily concentrations and thus better revealed trends. Finally, the daily average mixing ratios of $O_x$ (= $O_3$ + $NO_2$) were calculated to account for the titration effect of nitric oxide (NO) on changes in $O_3$ concentration.

## 2.2 Satellite and meteorological data

Satellite data were used to investigate the spatiotemporal variations in tropospheric formaldehyde (HCHO) and $NO_2$ column concentrations during the 2022 LCD period and the same period in 2021. We obtained Sentinel-5P Level-3 Offline products (HCHO and $NO_2$) using the TROPOspheric Monitoring Instrument (TROPOMI) from the Google Earth Engine (GEE; https://earthengine.google.com/) cloud-based platform, which is an open-source processing system based on JavaScript (Ghasempour et al., 2021). The Sentinel-5P Level-3 Offline products were converted from the original Sentinel-5P Level-2 Offline data (at a resolution of 5.5 × 3.5 km) by GEE using the harpconvert tool (https://cdn.rawgit.com/stcorp/harp/master/doc/html/harpconvert.html) and subjected to a data quality control process, in which pixels with data quality values less than 75% for $NO_2$ and less than 50% for HCHO were removed. Based on the GEE platform, we imported the daily image collections of HCHO and $NO_2$, cut them in accordance with the administrative boundary shapefiles of Shanghai (generated from publicly available geoscience data on the DataV.GeoAtlas platform; http://datav.aliyun.com/portal/school/atlas/area_generator) using a .*clip()* script (Gorelick et al., 2017) and then averaged the values of all the remaining pixels within the city border to obtain the daily mean concentrations of HCHO and $NO_2$. A similar satellite data processing method was used for the YRD region.

We acquired surface site meteorological data (2-m temperature, 2-m relative humidity [RH], 10-m wind direction and 10-m wind speed) from 78 weather stations in Shanghai during April–May in 2021 and 2022, together with gridded meteorological data (2-m temperature, 1,000-hPa RH, downward ultraviolet (UV)

radiation at the surface, total cloud cover, total precipitation and boundary layer height) from European

Centre for Medium-Range Weather Forecast Reanalysis v5 (ERA5)
(https://cds.climate.copernicus.eu/cdsapp#!/dataset/reanalysis-era5-single-levels and
https://cds.climate.copernicus.eu/cdsapp#!/dataset/reanalysis-era5-pressure-levels). We calculated monthly

averages of the surface-observed meteorological data in 2021 and 2022 and the changes in gridded

meteorological data between 2022 and 2021 to determine the spatiotemporal variations in meteorological

conditions in Shanghai and its surrounding regions.

**2.3 Backward trajectories analysis**

The 24-h backward trajectories for Shanghai were calculated at 1-h intervals during April–May 2022 using

MeteoInfoMap software (Wang, 2014, 2019) and meteorological data from the Global Data Assimilation

System (ftp://arlftp.arlhq.noaa.gov/pub/archives/gdas1/). The endpoints of the trajectories were 500 m above

ground level of Shanghai (31.23°N, 121.47°E). Cluster classification was then conducted to divide the

trajectories of $O_3$ exceedance days into five groups based on the origins of air mass.

**2.4 Site classification and regression analysis**

We classified the aforementioned 19 environmental monitoring sites in Shanghai into four types based on

land use and mean $HCHO/NO_2$ ratios (Fig. 1 and Fig. 8b). The type A sites (12 sites) are in the city centre

(and almost all are within the Shanghai Outer Ring Expressway) and had the lowest $HCHO/NO_2$ ratios; type

B are sites in the city perimeter (4 sites) that had moderate $HCHO/NO_2$ ratios; type C are semi-rural sites (2

sites) that had the largest $HCHO/NO_2$ ratios; and type D site (one site) is located near the wetland park in the

East Chongming Tidal Flat and is least affected by urban emissions but may be influenced by ship emissions.

Based on the above classification, we conducted regression analyses between the surface MDA8 $O_3$

concentrations and daily mean $NO_2$ concentrations for three types of sites (A, B and C) to examine the $O_3$

formation regime during the 2022 LCD in Shanghai's city centre, city perimeter and semi-rural sites.

**3 Results and discussion**

**3.1 Spatiotemporal variations in surface MDA8 $O_3$ and $NO_2$ concentrations**

Figure 2 presents the MDA8 $O_3$ and daily average $NO_2$ concentrations for the first 5 months of 2021–2022 at the 19 sites in Shanghai. In April–May 2022, the concentrations of $NO_2$, derived largely from fossil-fuel combustion, decreased by 49% year-on-year, but the average MDA8 $O_3$ concentrations increased by 12.5%. (In contrast, $PM_{2.5}$ concentrations fell by 30.3% in Shanghai, Fig. S2b). The MDA8 $O_3$ concentrations exceeded 160 μg/m³, which is China's Ambient Air Quality Standard for $O_3$, at one or more of the 19 sites on 21 days in April–May 2022, with the highest value being 200 μg/m³. The average $O_x$ ($O_3 + NO_2$) mixing ratio in April–May 2022 also increased by 2.2% during the LCD compared with the same period in 2021 (Fig. S2a). These observations indicate that there was significant ground-level $O_3$ pollution in Shanghai during the 2022 LCD, despite the drastic reductions in human activity. The decrease in $NO_2$ concentrations (49%) during the 2022 LCD was larger than the $NO_2$ decrease during the full-scale LCD phase in 2020 (23 January–12 February), when the average $NO_2$ concentration decreased by 30.5% from the corresponding period in 2019 (Fig. S3b). During the 2020 LCD, the average MDA8 $O_3$ concentrations in Shanghai increased by 13.8% but remained far below the ambient $O_3$ concentration standard (Fig. S3a).

Figure 3a-c shows the spatial distribution of the MDA8 $O_3$ concentrations in mainland China and the YRD region, which consists of Shanghai and three neighbouring provinces (Jiangsu (JS), Anhui (AH) and Zhejiang (ZJ)), and Shanghai during the 2022 LCD, respectively. It can be seen that elevated $O_3$ concentrations were observed in most sites in JS, in the southern part of ZJ and in the eastern part of AH. These provinces are known to have close economic ties to Shanghai.

The increase in MDA8 $O_3$ concentrations during the 2022 LCD was not restricted to Shanghai. Figure 3d-f shows the spatial distribution of changes in MDA8 $O_3$ concentrations from 2021 to 2022 at the national, regional and city scales, respectively. The MDA8 $O_3$ concentrations clearly increased in central and eastern regions but decreased in southwestern, northwestern and northeastern regions (Fig. 3d). The YRD region recorded one of the greatest increases in MDA8 $O_3$ concentrations (17.4 μg/m³) during the 2022 LCD (Fig.

3e). Within Shanghai, the year-on-year increase in MDA8 $O_3$ concentrations decreased from urban to semi-rural areas, with the largest increase recorded in the Putuo district in the city centre (32.9 µg/m$^3$) (Fig. 3f).

**3.2 TROPOMI-based HCHO and NO₂ concentrations**

Figure 4 depicts the spatial distributions of TROPOMI-based tropospheric column concentrations of HCHO and NO₂ and the HCHO/NO₂ column ratios in the YRD region. During the 2022 LCD and compared with the same period in 2021, the column concentrations of HCHO, derived from both direct emissions and degradation of anthropogenic and natural volatile organic compounds (VOCs), decreased by 10.8% (Fig. 4a), and the NO₂ column concentrations decreased by 25.1% (Fig. 4b). The magnitudes of the reductions in HCHO and NO₂ column concentrations had similar spatial distributions, with the largest reductions observed in the region's major cities (Shanghai, Suzhou, Nanjing and Hangzhou). This indicated that the reductions in both the NO₂ and HCHO column concentrations were related to decreases in anthropogenic activity during the 2022 LCD. Due to a larger decrease in NO₂ concentrations than in HCHO concentrations, the average HCHO/NO₂ ratio increased from 2.57 in 2021 to 2.96 in 2022 in the YRD region (Fig. 4c). In Shanghai, the HCHO and NO₂ column concentrations decreased by 19.4% and 49.2%, respectively, during the 2022 LCD (see Section 3.3.2 for a detailed discussion). In comparison, the TROPOMI-based HCHO and NO₂ column concentrations in February 2020 showed 17% and 38% decreases, respectively, in the YRD region compared with the same month in 2019 (Stavrakou et al., 2021). The much larger reduction in NO₂ concentrations than in HCHO concentrations during the 2022 LCD is attributable to the fact that $NO_x$ is mainly emitted by transportation activities (which exhibited the greatest decrease during the 2022 LCD) and by power generation, whereas VOCs are derived from more diverse sources. In Shanghai, the main VOC sources are vehicular exhaust, evaporation of fuels, paints and solvents, petrochemical industries, liquefied petroleum gas and biogenic sources (Lin et al., 2020; Han et al., 2022).

### 3.3 Cause(s) of high $O_3$ concentrations during the 2022 LCD

### 3.3.1 Effects of meteorological conditions on $O_3$ pollution in Shanghai

A large body of literature has indicated that meteorological conditions can significantly affect $O_3$ concentrations by altering emissions, chemical reaction rates, and distribution and removal processes (e.g., Lu et al., 2019; Liu and Wang, 2020; Liu et al., 2021; Jacob and Winner, 2009; Lin et al., 2008; He et al., 2017). To gain insights into the weather conditions during the 2022 LCD period, we compared several meteorological parameters recorded at surfaces in Shanghai (temperature, RH, and wind speed and direction)

(Fig. 5) and the ERA5 reanalysis data for large regions (Fig. 6). The results indicate that during the 2022 LCD, the average surface air temperature and RH decreased compared with the same period in 2021. That is, the mean temperature was 18.4 °C in 2022 versus 19.9 °C in 2021, and the mean RH was 71.4% in 2022 versus 74.2% in 2021 (Fig. 5a-b). The ERA5 reanalysis data revealed there was a decrease in the surface (2-m) temperature in Shanghai, whereas there was an obvious increase in the 2-m temperature in the northern

part of the YRD region and a large decrease in the 2-m temperature the southern part of the YRD region (Fig. 6). The decrease in mean temperature in Shanghai during the 2022 LCD may have slowed $O_3$ production by reducing chemical reaction rates and biogenic emissions. The ERA5 reanalysis data also revealed that during the 2022 LCD there were insignificant changes in cloud cover, downward UV radiation, boundary layer heights and total precipitation in Shanghai but considerable changes in these parameters in the surrounding

areas (Fig. 6).

We also assessed whether there was a significant change in surface air flow in Shanghai during the 2022 LCD compared with the preceding year. An examination of surface winds in Shanghai showed that during the 2022 LCD, predominant surface winds were from the north–east–southeast sectors (Fig. 5c-d), which is consistent with the 24-h calculated back trajectories (Fig. S4). This indicates that for the majority of the 2022

LCD, Shanghai was upwind of other cities in the YRD region. Compared with the same period in 2021, during the 2022 LCD there was an increase in the occurrence of northerly winds and a decrease in the

occurrence of westerly winds. To check for transport of $O_3$ from high-$O_3$ areas in the northwest direction, we examined surface wind flows during the 21 $O_3$-exceedance days. Both surface wind flows and back trajectories indicated that during these days air mainly came from the northeast–east–southeast directions (Fig. 7), indicating that air from other YRD cities contributed little to the high-$O_3$ days in Shanghai during the 2022 LCD.

The above results suggest that the increase in $O_3$ concentrations in Shanghai during the 2022 LCD were not due to changes in meteorological conditions but were a result of local chemical production.

### 3.3.2 Effect of the changes in $O_3$ formation regimes in Shanghai

The photochemical production of $O_3$ is controlled by the non-linear chemistry of $NO_x$ (NO and $NO_2$) and VOCs (NRC, 1992). It is well known that in many cities, $O_3$ concentrations decrease as VOC emissions decrease but increase as $NO_x$ emissions decrease (e.g., Wang et al., 2022). The literature has shown that Shanghai largely operates within a VOC-limited regime (e.g., Lin et al., 2020). We found observational evidence – the HCHO/$NO_2$ ratios and the relationship between $O_3$ and $NO_2$ concentrations – that shows that the city remained in a VOC-limited regime before the 2022 LCD (both in the earlier months of 2022 and in April–May of 2021) but may have transitioned near a $NO_x$-VOC co-limited regime during the 2022 LCD. The HCHO/$NO_2$ ratio has been used as a proxy for $O_3$ formation regimes, with low ratios indicating VOC-limited conditions, high ratios indicating $NO_x$-limited conditions, and intermediate values indicating a co-limited regime. Figure 8 shows the temporal variations in city-average HCHO/$NO_2$ ratios in January–May of 2021 and 2022 (Fig. 8a) and the spatial variation of the ratios in Shanghai (Fig. 8b). It indicates that the ratios were comparable (0.56 versus 0.58) in the pre-2022-LCD months of 2021 and 2022 but significantly increased during the 2022 LCD (from 1.11 to 1.68) (Fig. 8a and Fig. S5). Moreover, the ratios were higher in the southern part than the northern part Shanghai, which houses the major urban districts (Fig. 8b).

Previous analysis of the historical OMI-HCHO/$NO_2$ ratios in Shanghai indicated a general rising trend in the ratios with values ranging from 0.5-1.5 in 2005-2019 in warm seasons (April-September) (Itahashi et al.,

2022; Li et al., 2021; Lee et al., 2022). For April and May, the months when the 2022 Shanghai LCD took place, the HCHO/NO$_2$ ratios ranged from 0.73 to 1.36 were in 9 out of 10 years during 2010-2019 with a higher ratio (1.61) for year 2014 (Li et al., 2021). This shows that the HCHO/NO$_2$ ratio during LCD 2022 (1.68) was high compared with that of the same months in the past decade, resulting from the sharply reduced

NO$_2$ column concentrations during the LCD. Several other megacities in East Asia have also seen increasing HCHO/NO$_2$ ratios during 2015-2019 (Itahashi et al., 2022; Lee et al., 2022).

To determine the regime transition threshold, we adopted an observation-based method similar to that which has been used by previous researchers (Jin et al., 2020; Wang et al., 2021; Schroeder et al., 2022): we plotted the city-average MDA8 O$_3$ concentrations against the HCHO/NO$_2$ ratio for the first 5 months of 2021 and

225 2022 (Fig. 9). The peak O$_3$ concentrations increased as the HCHO/NO$_2$ ratio increased and plateaued at approximately 2, indicating that an HCHO/NO$_2$ ratio of 2 was the threshold for transition from a VOC-limited to a co-limited regime. This value is similar to that determined (2.3) for other major Chinese cities (Wang et al., 2021); however, it is less than that determined (~3) for several US cities (Jin et al., 2020) and greater than that which has been derived (1) from model simulations (Duncan et al., 2010; Li et al., 2021). The city-

230 averaged HCHO/NO$_2$ ratio was greater than 2 on 15 days during the 2022 LCD but on just 2 days in 2021 (see Fig. 9). Moreover, a spatial analysis shows that during the 2022 LCD, the southern part of Shanghai was in a VOC–NO$_x$ co-limited regime (with an HCHO/NO$_2$ ratio > 2), whereas the city centre in the northern part remained in the VOC-limited regime (Fig. 8b).

The relationship between surface O$_3$ and NO$_2$ concentrations supports the inference made based on satellite-

235 derived HCHO/NO$_2$ ratio data. It is known that O$_3$ concentrations are negatively correlated with NO$_x$, NO$_y$ or NO$_z$ concentrations (NO$_y$ = NO$_x$ + oxidation products of NO$_x$ [NO$_z$]) under a NO$_x$-titrated condition; this relationship is slightly positive (with small $\Delta O_3/\Delta NO_x$ ratios) in a VOC-limited regime and very positive (with large $\Delta O_3/\Delta NOx$ ratios) in a NO$_x$-limited regime. For example, previous studies have suggested that an afternoon $\Delta O_3/\Delta NO_z$ ratio (ppb/ppb) less than 4 corresponds to a VOC-limited regime, an afternoon

$\Delta O_3/\Delta NO_z$ ratio greater than 7 corresponds to a NO$_x$-limited regime and an afternoon $\Delta O_3/\Delta NO_z$ of 4–7

corresponds to a transition regime (Wang et al., 2017). Only $NO_2$ or $NO_x$ (not other forms reactive nitrogen) are measured by most regular air-monitoring networks, including the China Environmental Monitoring Network used in this study. Figure 10 shows the scatter plot of the MDA8 $O_3$ and daily average $NO_2$ concentrations in three types of sites in Shanghai: the city centre (12 sites), the city perimeter (4 sites) and the semi-rural areas (2 sites). The VOC-limited regime in the city centre sites (with an average $HCHO/NO_2$ ratio of 1.43) had the smallest $O_3/NO_2$ slope (3.4) and that in the transition regime sites (with an $HCHO/NO_2$ ratio of 2.27) had the largest $O_3/NO_2$ slope (5). In comparison, in 2021, the peak $O_3$ concentration had either a very weak or no correlation with $NO_2$ concentration**s**, indicative of a typical VOC-limited regime. (The type D site had small positive $O_3/NO_2$ slopes in both years (0.89-1.17), with high $NO_2$ during the 2022 LCD possibly from the ship emissions in the Yangtze River Estuary, Fig. S6.)

The above results suggest that the increased $O_3$ concentrations observed during the 2022 LCD were mainly due to increased $O_3$ production, which resulted from a larger reduction in $NO_x$ emissions than in VOC emissions under a VOC-limited condition. Additionally, the decrease in particulate emissions, which reduces the uptake of radicals and $O_3$ and increases radiation, could have increased $O_3$ concentrations. Previous model simulations of the 2020 LCD in central China, which saw similar decreases in emissions (i.e., $NO_x$: ~50%, PM: ~30%), showed that the decrease in $NO_x$ emissions made a much larger contribution than the decrease in PM emissions to the increase in $O_3$ concentrations (Liu et al., 2021). We believe that this also occurred during the 2022 LCD, i.e., the $O_3$ concentration increase was mainly due to enhanced $O_3$ production, which occurred as a result of a large reduction in $NO_x$ emissions. The $O_3$ formation regime during the 2022 LCD remained VOC-limited in urban areas but entered co-limiting conditions outside the city centre.

**4 Summary and implications**

This study analysed the causes of frequent ground-level $O_3$ pollution during the 2022 LCD (April–May 2022) in Shanghai and assessed the increases in $O_3$ concentrations compared with the same periods in previous years using ground and satellite-based observations. We found that despite large reductions in the activities

of transportation sectors and industries during the 2022 LCD, frequent exceedances of the $O_3$ air quality

standard (~30% days) were observed at ground level. Moreover, the $O_3$ concentrations were increased during

the 2022 LCD compared with the same period in the preceding year. This increase resulted from a large

reduction in $NO_x$ emissions (~50% from surface and satellite-based measurements) and a small reduction in

VOCs (19% in HCHO column concentrations) within Shanghai. In contrast, meteorology and outside

influences had insignificant effects on $O_3$ concentrations during the 2022 LCD. Moreover, $O_3$ formation

during the 2022 LCD remained in the VOC-limited regime at most urban and suburban sites but transitioned

to a VOC–$NO_x$ co-limited regime in semi-rural areas.

Our findings on the $O_3$ response to the 2022 LCD have implications for mitigating summer $O_3$ pollution,

which has become the predominant air-pollution problem during warm seasons in China. $O_3$ pollution is also

a persistent environmental hazard in the US and Europe even after a few decades of research and control (e.g.,

Tao et al., 2022; Derwent and Parrish, 2022). Similar to Shanghai, many of the world's cities, such as Los

Angeles and New York, are still in VOC-limited regimes (Jin et al., 2020; Tao et al., 2022). Our results show

that drastically decreasing emissions from conventional fossil fuel-powered transportation sectors during the

2022 LCD can lead to increased and elevated $O_3$ concentrations in VOC-limited urban areas. China is aiming

to have its carbon emissions peak by 2030 and to achieve carbon neutrality by 2060, and other countries are

also committing to drastically reduce their carbon emissions. This will necessitate the rapid uptake of electric

vehicles in many cities. However, although large-scale adoption of electric vehicles will greatly improve

overall air quality, VOC emissions from other sectors will need to be decreased at the same time to prevent

increases in $O_3$ concentrations from the reductions in $NO_x$ (and particulate) emissions in the early stages of

phase-out of conventional vehicles. Over time, the wide application and acceptance of renewable energy in

transportation and energy production will help cities reach the $NO_x$-limited formation regime, alleviate

ground-level $O_3$ pollution and achieve carbon-reduction targets. The aggravated $O_3$ pollution during the 2022

Shanghai LCD after large reductions in transportation emissions suggests that it may take considerably long

time for some cities reach a $NO_x$-limited regime.

*Code and data availability.* The code or data used in this study are available upon request from Tao Wang (tao.wang@polyu.edu.hk).

*Supplement.* The supplement related to this article is available online at:

*Author contributions.* TW initiated the research and designed the framework of data analysis. YT processed the data and made the figures. TW and YT analysed the results and wrote the paper.

*Competing interests.* One author (Tao Wang) is a member of the editorial board of Atmospheric Chemistry and Physics. The peer-review process was guided by an independent editor, and the authors have no other competing interests to declare.

*Acknowledgements.* This research has been supported by the Hong Kong Research Grants Council (grant no. T24-504/17-N) and the National Natural Science Foundation of China (grant no. 91844301). We thank

Shanshan Wang at Fudan University for providing the OMI-HCHO/$NO_2$ ratio data in Shanghai during 2010-2019.

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

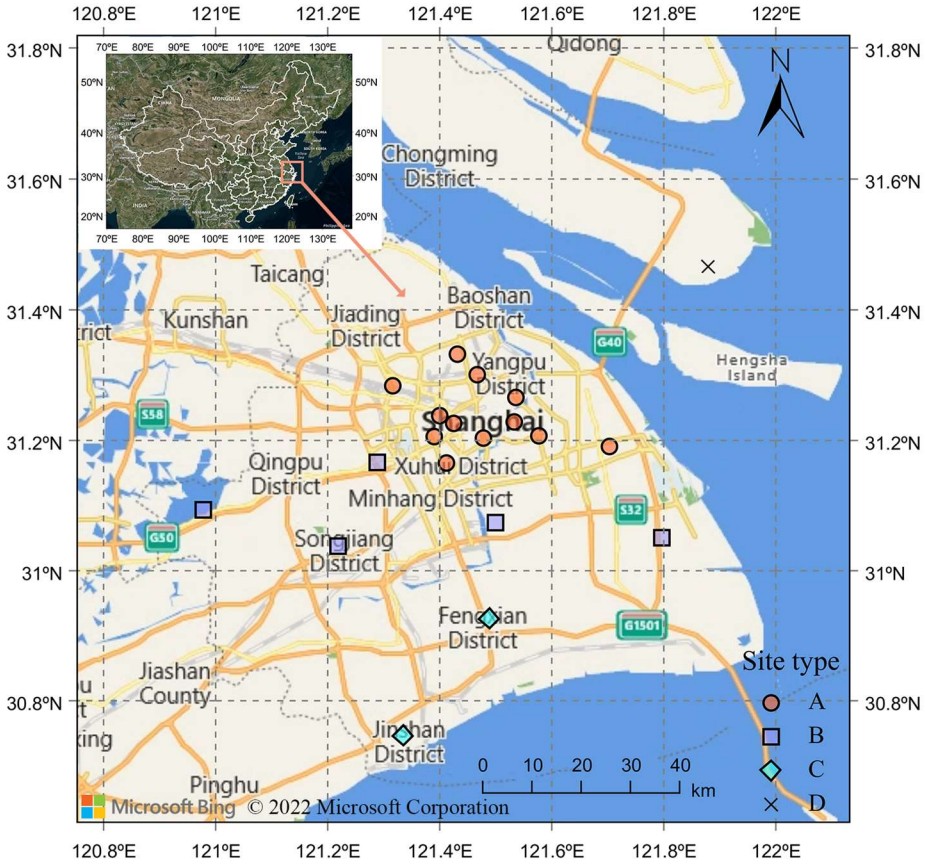

**Figure 1.** Spatial distribution and classification of the 20 environmental monitoring sites in Shanghai. Types A, B, C and D represent sites in the city centre (12 sites, circles), city perimeter (5 sites, squares), semi-rural area (2 sites, diamonds), and the Yangtze River Estuary (one site, cross), respectively.

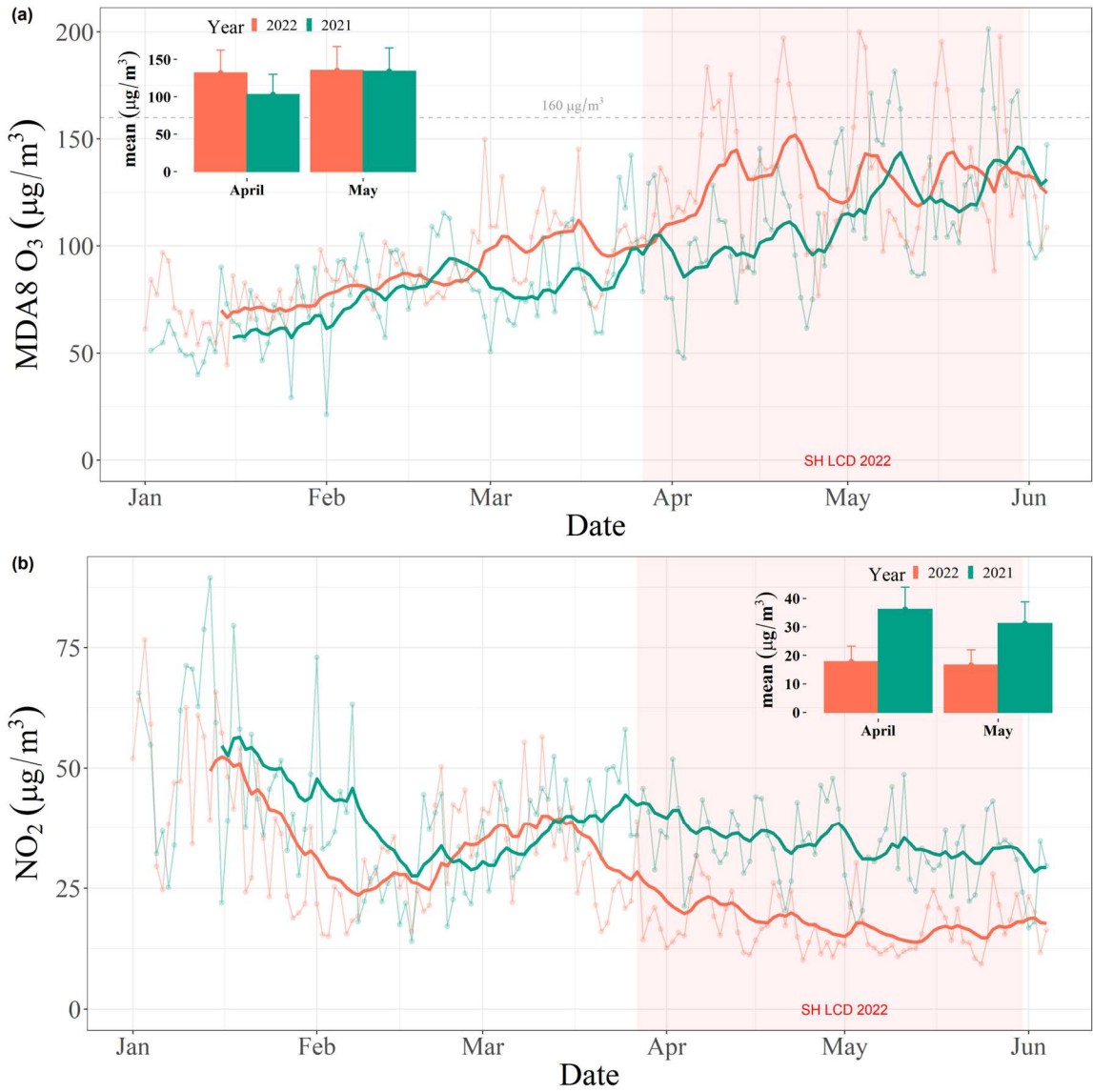

**Figure 2.** Temporal variations in surface-level maximum daily 8-h average (MDA8) $O_3$ (a) and $NO_2$ (b) concentrations during the 2022 pre-lockdown (LCD) (1 January to 27 March) and LCD periods (28 March to 31 May) compared with the corresponding periods in 2021. The concentrations are averages from 19 sites in Shanghai. The trend lines represent prior 2-week moving averages. The insert figures show the monthly average concentrations (bars and values indicated by numbers) and standard deviations (error bars). The red background represents the 2022 LCD in Shanghai (SH LCD 2022: 28 March to 31 May). The *x*-axis labels represent the first day of each month.

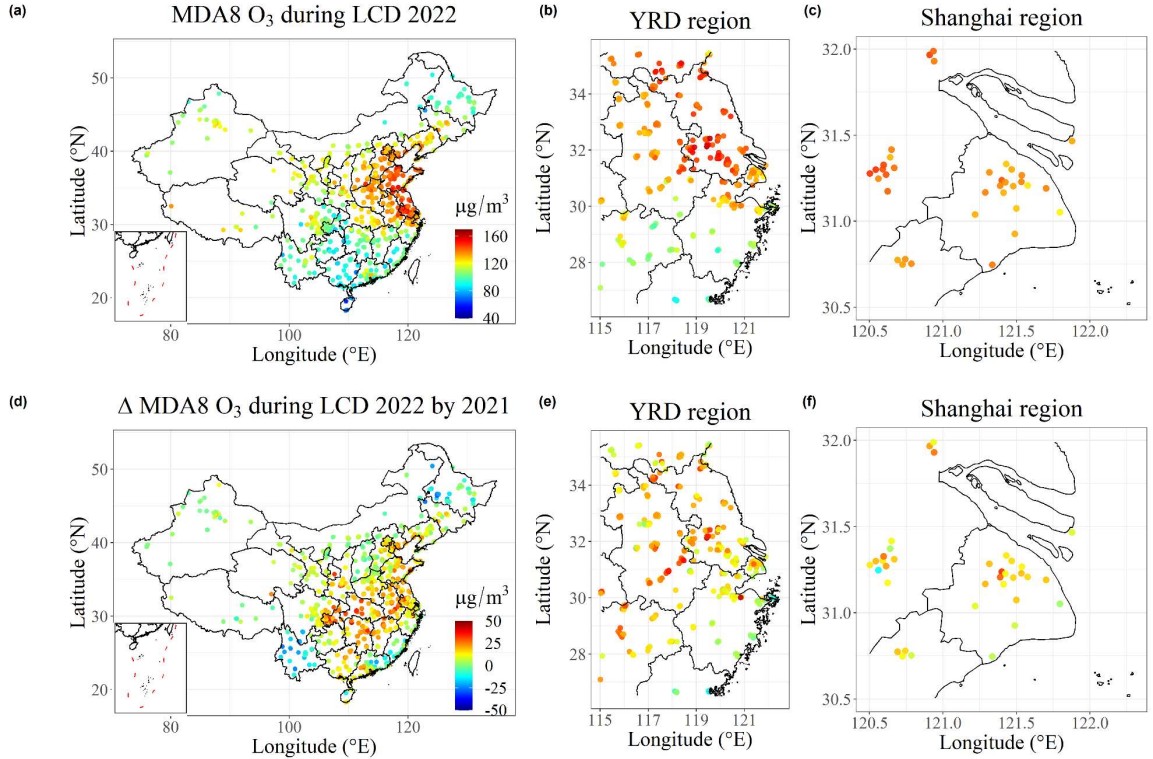

**Figure 3.** Spatial distribution of surface maximum daily 8-h average (MDA8) O₃ concentrations in 2022 and changes in the same compared with 2021 across all of mainland China (a and d), in the Yangtze River Delta region (b and e) and in Shanghai (c and f) from 28 March to 31 May 2022. LCD: lockdown.

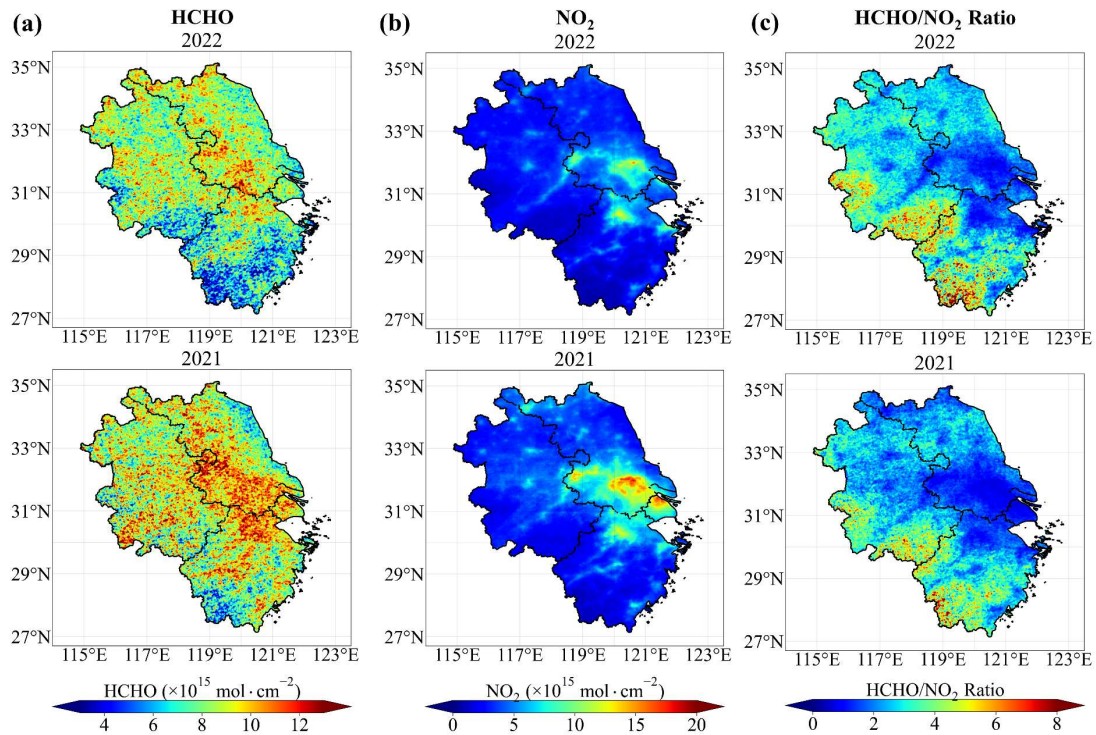

**Figure 4.** Spatial distribution of the TROPOspheric Monitoring Instrument-based tropospheric column concentrations of HCHO (a) and NO₂ (b) and the HCHO/NO₂ ratio (c) in the Yangtze River Delta region during the 2022 lockdown (upper row) and compared with the same period in 2021 (lower row).

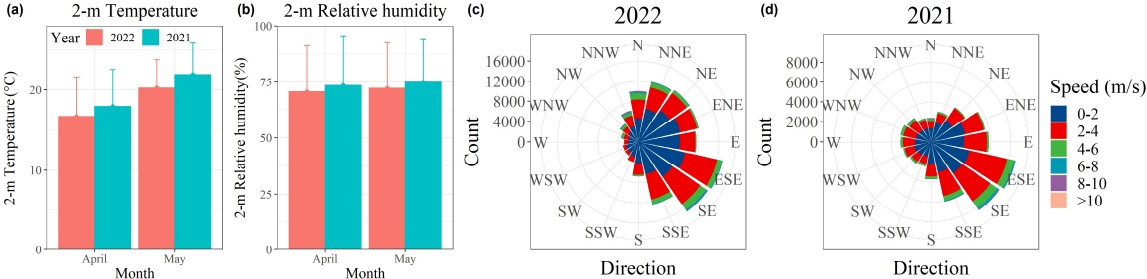

**Figure 5.** A comparison of average temperature (a) and relative humidity (b) in April and May of 2021 and 2022. Wind rose plots for the 2022 lockdown period (c) and the same period in 2021 (d). The data are surface observations.

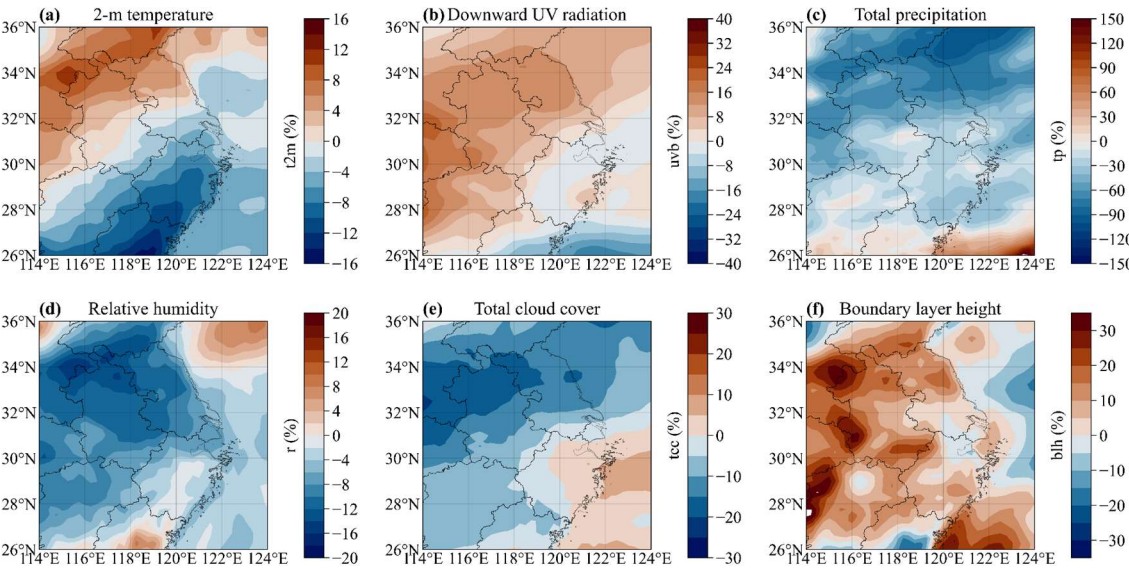

**Figure 6.** Percentage change in the average meteorological parameters of European Centre for Medium-Range Weather Forecast Reanalysis v5 data for the Yangtze River Delta and its neighbouring regions. 2-m temperature (t2m) (a), downward ultraviolet (UV) radiation at the surface (uvb) (b), total precipitation (tp) (c), relative humidity at 1,000 hPa (r) (d), total cloud cover (tcc) (e) and boundary layer height (blh) (f).

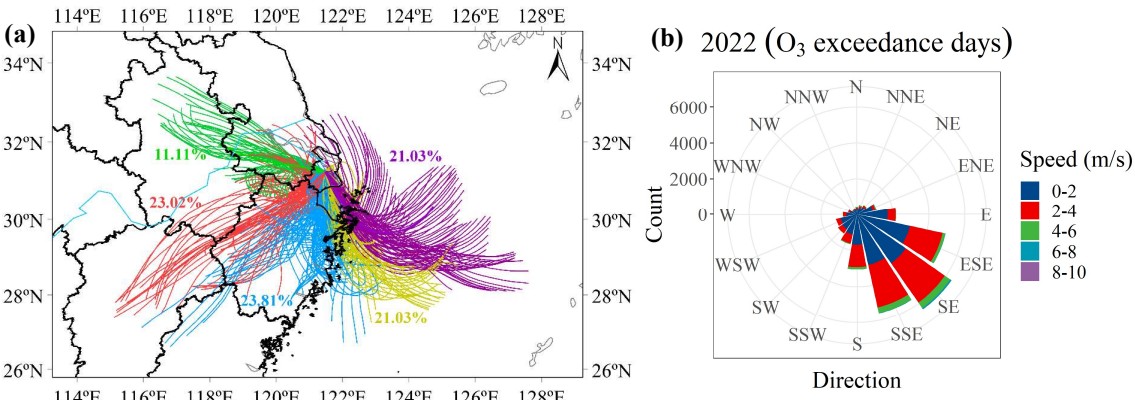

**Figure 7.** (a) 24-h back trajectories at 1-h intervals at a height of 500 m; and (b) surface wind rose plot for the 21 O$_3$ exceedance days during the 2022 lockdown in Shanghai. The inserted percentages represent the percentages of trajectory numbers of each classification in the whole trajectories.

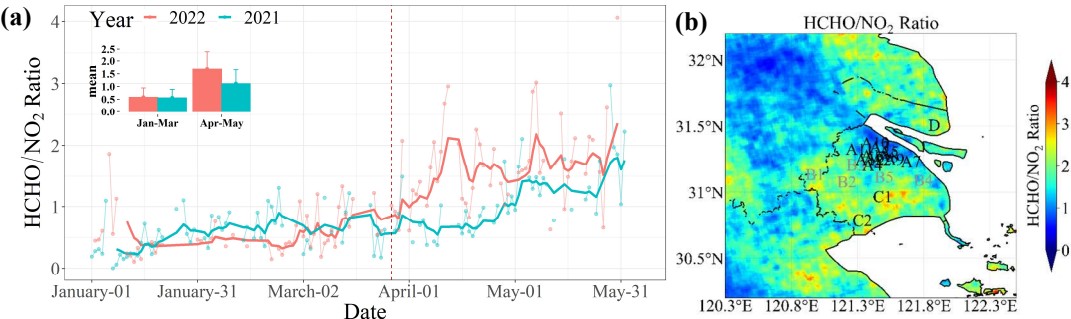

**Figure 8.** Temporal variations in the city-average HCHO/NO$_2$ ratio from 1 January to 31 May of 2021 and 2022 (a). Spatial variations in the HCHO/NO$_2$ ratio in Shanghai and its surrounding regions during 2022 lockdown (b). The trend lines in (a) represent the moving average by 2 weeks. The red dashed vertical line represents the lockdown starting date (27 March) and the insert shows the monthly average HCHO/NO$_2$ ratios with standard deviations.

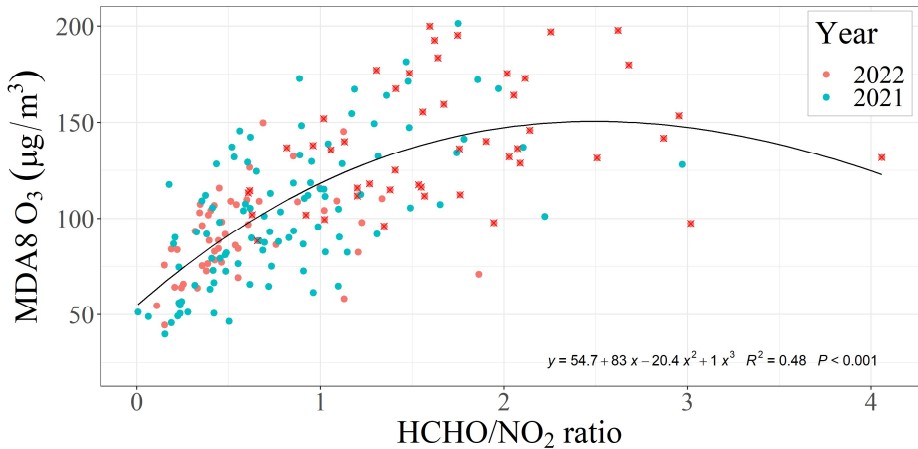

**Figure 9.** Scatter plot of Shanghai's average maximum daily 8-h (MDA8) O$_3$ concentrations and HCHO/NO$_2$ ratios in January–May of 2021 (blue dots) and 2022 (red dots). The data points recorded during the lockdown periods are labelled with red crosses.

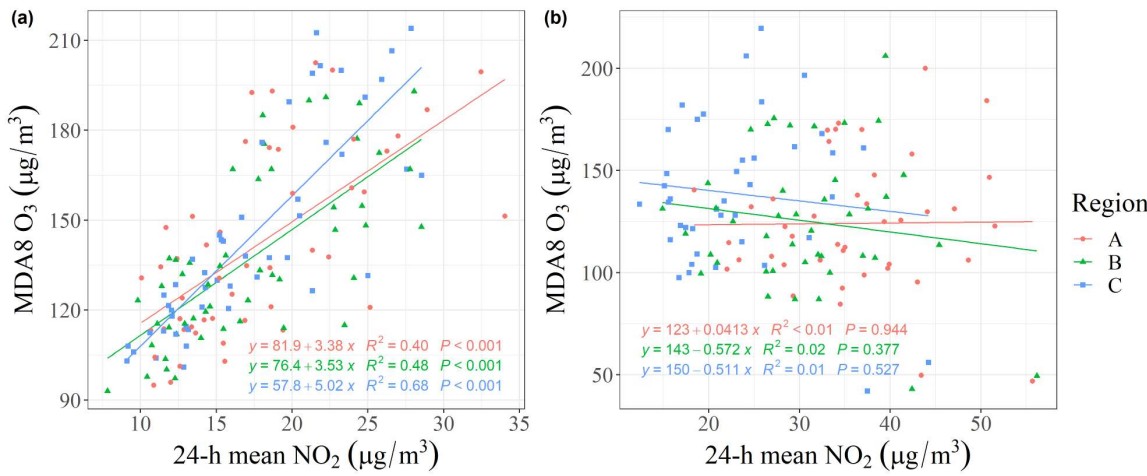

**Figure 10.** Scatter plots of the surface maximum daily 8-h average (MDA8) $O_3$ concentrations versus surface daily $NO_2$ concentrations and their linear regressions (straight lines) during the 2022 lockdown (a) and the same period in 2021 (b) for 3 types of sites.