# Peer review of "What caused ozone pollution during the 2022 Shanghai lockdown? Insights from ground and satellite observations"

_EGUsphere, 2022_

## Referee Comment (RC2)

This study describes a study in 2022 Shanghai lockdown by investigating the spatiotemporal variations in surface MDA8 $O_3$ and $NO_2$ concentrations, TROPOMI-based HCHO and $NO_2$ concentrations, and the cause of high $O_3$ concentrations during the 2022 LCD from two aspects: meteorological conditions and $O_3$ formation regimes. It have great insight on $O_3$ concentration increased in 2022 LCD, having greatly detailed data and substantial content. I am glad that the author can report the recent ozone pollution situation to us in a short time and make an in-depth analysis. But there are some problems, which must be solved before it is considered for publication. Below I have reported the lines in the text that require further investigation and clarification.

1. L25: the abbreviations of nitrogen should be $NO_x$ rather than NOx. Elsewhere the abbreviation is correct

2. 2.1 section: There is a lack of description of contaminant monitoring instruments and their detection limits.

3. L63: 3.3.1 title "Effects of meteorological conditions on $O_3$ concentrations in Shanghai", I suggest changing the "$O_3$ concentrations" to "$O_3$ pollution".

4. L164: "A large body of literature has indicated that meteorological conditions can significantly affect $O_3$ concentrations ⋯". there are only two references in the back, which seems a little weak. Suggest adding more references.

5. Fig.2: the inset figures show the monthly average concentrations, but the y-axis is missing units.

6. L126: based on the coverage in Shanghai that 2020 LCD date from 23 Jan to 23 Mar, but in this paper is 23 Jan to 12 Feb. Hope you can reconfirm it.

7. L209 (Fig.9): in this picture, your x-axis is HCHO/$NO_2$ ratio, and y-axis is MDA8 $O_3$ (ug/$m^3$), but OMI HCHO/$NO_2$ and high-$O_3$ Probability (%) in your reference paper, respectively (Wang et al., 2021). Can you explain this difference? Weather the change of axis will affect $O_3$ formation regimes?

---

## Author Comment (AC1)

We would like to thank both reviewers for their helpful, detailed, and constructive comments. We can include all suggestions in the following revision of the manuscript.

Reviewers' comments are shown in black text. Then we use blue and red text to denote our responses to the comments and the corresponding edits, respectively.

**Response to Reviewer #1**

We thank the reviewer for the comments and suggestions on the manuscript. Our response (in blue) and the corresponding edits (in red) are shown below.

This study is well done and the paper clear and concise. However, the focus is rather local.

One recommendation is to put the findings of the paper into better context. For instance, the NO titration effect is well documented in the literature over the last 30+ years in the US and Europe and more recently in China. So, is it a surprise that you see this effect in Shanghai and other large Chinese cities? Not really as you indicate. Other large world cities have struggled with the issue of decreasing emissions enough to begin seeing reductions in ozone. It would be worth discussing the literature on this topic.

It would also be helpful to discuss better (a few paragraphs) on how $HCHO/NO_2$ and ozone trends have evolved over the satellite record. You only looked at TROPOMI, but the OMI data record is much longer (2005-present). How has the $HCHO/NO_2$ evolved over time in Shanghai and other Chinese cities? This analysis would help to put the magnitude of the 2022 LCD into historical context and help policy-makers to know what VOC and NOx emissions reductions will be required to see ozone decrease. There are studies in the literature that have looked at the historical $HCHO/NO_2$ in China, so you may not need to do your own analysis.

These two recommendations will help you to more broadly discuss the implications of your work for other Chinese cities and to better put the importance of your work (e.g., Summary and Implications) into context. I'm recommending this since your local focus on Shanghai may not fit the aims of ACP: "The journal scope is focused on studies with important implications for our understanding of the state and behavior of the atmosphere. Articles with a local focus must clearly explain how the results extend and compare with current knowledge."

Response: We thank the referee for the valuable suggestions.

We agree with the suggestion for more discussion on the general implication of this study. We used the recent lockdown in Shanghai as a case to disentangle the roles of meteorology and chemistry in the occurrence of $O_3$ pollution after massive reductions in emissions in a megacity and the surrounding regions. Compared with the 2020 lockdown, the 2022 lockdown occurred in a warm season and lasted much longer, thereby serving as a relevant real-world test of the $O_3$ response to emission reductions in a high-$O_3$ season." It is of particular interest to know whether the huge decreases in NOx emissions during LCD could push a typical VOC-limited megacity to a NOx-limited regime, which would offer guidance for future $O_3$ and NOx control. We show that even after 70%-85% decrease in transportation activities, urban areas of Shanghai are still in the VOC-limited condition, and the result implies it would take a long way for Shanghai and other VOC-limited urban areas in China to reach to NOx control. Therefore, the findings of this study have are relevant to

ozone control for other cities.

We have added the following text to clarify the broader implications of this study:

"Our findings on the $O_3$ response to the 2022 LCD have implications for mitigating summer $O_3$ pollution, which has become the predominant air-pollution problem during warm seasons in China. $O_3$ pollution is also a lingering environmental hazard in the US and Europe even after a few decades of research and control (e.g., Tao et al., 2022; Derwent and Parrish, 2022). Similar to Shanghai, many of the world's cities, such as Los Angeles and New York, are still in VOC-limited regimes (Jin et al., 2020; Tao et al., 2022). Our results show that drastically decreasing emissions from conventional fossil fuel-powered transportation sectors during the 2022 LCD can lead to increased and elevated $O_3$ concentrations in VOC-limited urban areas. China is aiming to have its carbon emissions peak by 2030 and to achieve carbon neutrality by 2060, and other countries are also committing to drastically reduce their carbon emissions. This will necessitate the rapid uptake of electric vehicles in many cities. However, although large-scale adoption of electric vehicles will greatly improve overall air quality, VOC emissions from other sectors will need to be decreased at the same time to prevent increases in $O_3$ concentrations from the reductions in $NO_x$ (and particulate) emissions in the early stages of phase-out of conventional vehicles. Over time, the wide application and acceptance of renewable energy in transportation and energy production will help cities reach the $NO_x$-limited formation regime, alleviate ground-level $O_3$ pollution and achieve carbon-reduction targets. The aggravated $O_3$ pollution during the 2022 Shanghai LCD amid such large reductions in transportation emissions suggests that it may take considerably long time for some cities reach a $NO_x$-limited regime."

Following the referee's suggestion, we have added the historical OMI HCHO/$NO_2$ ratios from the literature. The result shows increasing ratios in the past decade, reaching the highest value during the 2022 LCD. The following text has been added:

"Previous analysis of the historical OMI-HCHO/$NO_2$ ratios in Shanghai indicated a general rising trend in the ratios with values ranging from 0.5-1.5 in 2005-2019 in warm seasons (April-September) (Itahashi et al., 2022; Li et al., 2021; Lee et al., 2022). For April and May, the months when the 2022 Shanghai LCD took place, the HCHO/$NO_2$ ratios ranged from 0.73 to 1.36 were in 9 out of 10 years during 2010-2019 with a higher ratio (1.61) was found for year 2014 (Li et al., 2021). This shows that the HCHO/$NO_2$ ratio during LCD 2022 (1.68) was extremely high compared with that of the same months in the past decade, resulting from the sharply reduced $NO_2$ column concentrations during LCD. Several other megacities in East Asia have also seen increasing HCHO/$NO_2$ ratios during 2015-2019 (Itahashi et al., 2022; Lee et al., 2022)."

According to the above modified content, we also modified the *Abstract* and the *Introduction* accordingly as follows:

"… During the 2022 lockdown, Shanghai's urban centre remained VOC-sensitive despite drastic reductions in road transportation (73%-85%) and industrial emissions (~40%), whereas its semi-rural areas transitioned from VOC-limited to VOC–$NO_x$ co-limited regimes. Our findings suggest that future emission reductions similar to those that occurred during the lockdown, such as those that will result from electrifying transportation, will not be sufficient to eliminate $O_3$ pollution in urban areas of Shanghai and possibly other VOC-limited metropolis, without the imposition of

additional VOC controls or more substantial decreases in $NO_x$ emissions. …"

"Compared with the 2020 LCD in China, which was imposed in late January–early February, the 2022 LCD in Shanghai was imposed in the high-$O_3$ season and lasted for longer, during which time the $O_3$ concentrations exceeding the air quality standard were frequently observed. In the present study, we aimed to understand how meteorology and non-linear chemistry influenced ground-level $O_3$ concentrations during the 2022 LCD and whether the huge decrease in $NO_x$ emission could push a typical VOC-limited megacity to a $NO_x$-limited regime."

Reference:

Derwent, R. G. and Parrish, D. D.: Analysis and assessment of the observed long-term changes over three decades in ground-level ozone across north-west Europe from 1989 - 2018, Atmospheric Environment, 286, 119222, https://doi.org/10.1016/j.atmosenv.2022.119222, 2022.

Itahashi, S., Irie, H., Shimadera, H., and Chatani, S.: Fifteen-Year Trends (2005–2019) in the Satellite-Derived Ozone-Sensitive Regime in East Asia: A Gradual Shift from VOC-Sensitive to NOx-Sensitive, Remote Sensing, 14, 4512, https://doi.org/10.3390/rs14184512, 2022.

Jin, X., Fiore, A., Boersma, K. F., Smedt, I. D., and Valin, L.: Inferring Changes in Summertime Surface Ozone–NOx–VOC Chemistry over U.S. Urban Areas from Two Decades of Satellite and Ground-Based Observations, Environ. Sci. Technol., 54, 6518–6529, https://doi.org/10.1021/acs.est.9b07785, 2020.

Lee, H.-J., Chang, L.-S., Jaffe, D. A., Bak, J., Liu, X., Abad, G. G., Jo, H.-Y., Jo, Y.-J., Lee, J.-B., Yang, G.-H., Kim, J.-M., and Kim, C.-H.: Satellite-Based Diagnosis and Numerical Verification of Ozone Formation Regimes over Nine Megacities in East Asia, Remote Sensing, 14, 1285, https://doi.org/10.3390/rs14051285, 2022.

Li, D., Wang, S., Xue, R., Zhu, J., Zhang, S., Sun, Z., and Zhou, B.: OMI-observed HCHO in Shanghai, China, during 2010–2019 and ozone sensitivity inferred by an improved HCHO / NO2 ratio, Atmospheric Chemistry and Physics, 21, 15447–15460, https://doi.org/10.5194/acp-21-15447-2021, 2021.

Tao, M., Fiore, A. M., Jin, X., Schiferl, L. D., Commane, R., Judd, L. M., Janz, S., Sullivan, J. T., Miller, P. J., Karambelas, A., Davis, S., Tzortziou, M., Valin, L., Whitehill, A., Civerolo, K., and Tian, Y.: Investigating Changes in Ozone Formation Chemistry during Summertime Pollution Events over the Northeastern United States, Environ. Sci. Technol., https://doi.org/10.1021/acs.est.2c02972, 2022.

**Line 222: "wug" is not a word. "with"?**
Response: Thanks for pointing it out. We have corrected the typo.

**Response to reviewer #2:**

We thank the reviewer for the comments and suggestions on our manuscript. Our response (in blue) and the corresponding edits (in red) are shown below.

**This study describes a study in 2022 Shanghai lockdown by investigating the spatiotemporal variations in surface MDA8 $O_3$ and $NO_2$ concentrations, TROPOMI-based HCHO and $NO_2$ concentrations, and the cause of high $O_3$ concentrations during the 2022 LCD from two aspects: meteorological conditions and $O_3$ formation regimes. It has great insight on $O_3$ concentration increased in 2022 LCD, having greatly detailed data and substantial content. I am glad that the author can report the recent ozone pollution situation to us in a short time and make an in-depth analysis. But there are some problems, which must be solved before it is considered for publication. Below I have reported the lines in the text that require further investigation and clarification.**

Response: We thank the referee for the supportive comments and valuable suggestions for changes. We have modified our manuscript according to these comments. Please refer to the following content.

**L25: the abbreviations of nitrogen should be $NO_x$ rather than NOx. Elsewhere the abbreviation is correct.**

Response: Thanks for the comment. We have changed the 'NOx' into '$NO_x$'.

**2.1 section: There is a lack of description of contaminant monitoring instruments and their detection limits.**

Response: Thanks for your suggestion. We have added a brief description of the automated monitoring systems as follows:

"…The ambient concentrations of $O_3$, $NO_2$ and $PM_{2.5}$ are measured by an automated monitoring system at each site and reported to the China National Environmental Monitoring Centre and published online after validation (Wang et al, 2014). $O_3$, $NO_2$, and $PM_{2.5}$ are measured with UV photometry, chemiluminescence, and micro-oscillating balance or β absorption, respectively, with a detection limit of 2 ppb, 2 ppb and 2 μg/m$^3$, respectively (https://www.mee.gov.cn). …"

Reference:

Wang, Y., Ying, Q., Hu, J., and Zhang, H.: Spatial and temporal variations of six criteria air pollutants in 31 provincial capital cities in China during 2013–2014, Environment International, 73, 413–422, https://doi.org/10.1016/j.envint.2014.08.016, 2014.

**L63: 3.3.1 title "Effects of meteorological conditions on $O_3$ concentrations in Shanghai", I suggest changing the "$O_3$ concentrations" to "$O_3$ pollution".**

Response: Thanks for your suggestion. We have changed the "$O_3$ concentrations" into "$O_3$ pollution".

**L164: "A large body of literature has indicated that meteorological conditions can significantly affect $O_3$ concentrations …". there are only two references in the back, which seems a little weak. Suggest adding more references.**

Response: Thanks for the suggestion. We have added four more references (Jacob and Winner, 2009;

He et al., 2017; Lin et al., 2008; Lu et al., 2019) to support the effects of meteorological conditions on $O_3$ concentrations and modified the sentence as follows:

"A large body of literature has indicated that meteorological conditions can significantly affect $O_3$ concentrations by altering emissions, chemical reaction rates, and distribution and removal processes (e.g., Lu et al., 2019; Liu and Wang, 2020; Liu et al., 2021; Jacob and Winner, 2009; Lin et al., 2008; He et al., 2017)"

Reference:

He, J., Gong, S., Yu, Y., Yu, L., Wu, L., Mao, H., Song, C., Zhao, S., Liu, H., Li, X., and Li, R.: Air pollution characteristics and their relation to meteorological conditions during 2014–2015 in major Chinese cities, Environmental Pollution, 223, 484–496, https://doi.org/10.1016/j.envpol.2017.01.050, 2017.

Lin, J.-T., Patten, K. O., Hayhoe, K., Liang, X.-Z., and Wuebbles, D. J.: Effects of Future Climate and Biogenic Emissions Changes on Surface Ozone over the United States and China, Journal of Applied Meteorology and Climatology, 47, 1888–1909, https://doi.org/10.1175/2007JAMC1681.1, 2008.

Lu, X., Zhang, L., and Shen, L.: Meteorology and Climate Influences on Tropospheric Ozone: a Review of Natural Sources, Chemistry, and Transport Patterns, Curr Pollution Rep, 5, 238–260, https://doi.org/10.1007/s40726-019-00118-3, 2019.

Jacob, D. J. and Winner, D. A.: Effect of climate change on air quality, Atmospheric Environment, 43, 51–63, https://doi.org/10.1016/j.atmosenv.2008.09.051, 2009.

**Fig.2: the inset figures show the monthly average concentrations, but the y-axis is missing units.**

Response: Thanks for pointing this out. The units of the y-axis in the inserted figures are the same as the units of the y-axis in the main figures. We have added the units of the inserted figures in Fig.2.

[Figure]

Figure 2. Temporal variations in surface-level maximum daily 8-h average (MDA8) O3 (a) and NO2

(b) concentrations during the 2022 pre-lockdown (LCD) (1 January to 27 March) and LCD periods (28 March to 31 May) compared with the corresponding periods in 2021. The concentrations are averages from 19 sites in Shanghai. The trend lines represent prior 2-week moving averages. The insert figures show the monthly average concentrations (bars and values indicated by numbers) and standard deviations (error bars). The red background represents the 2022 LCD in Shanghai (SH LCD 2022: 28 March to 31 May). The x-axis labels represent the first day of each month.

**L126: based on the coverage in Shanghai that 2020 LCD date from 23 Jan to 23 Mar, but in this paper is 23 Jan to 12 Feb. Hope you can reconfirm it.**
Response: Thanks for pointing this out.

The 23 Jan-12 Feb 2020 was the full scale LCD period at the national scale (Liu et al., 2021). During this period, Shanghai imposed the most stringent control, similar to the control during the two-month LCD in 2022. After the most stringent control phase in 2020, Shanghai relaxed some restrictions till 23 March 2020 (e.g., some essential workers returned to work while others continued to work from home, and essential travels (e.g, grocery shopping) were allowed but controlled) (Ding et al., 2021). To make a more meaningful comparison, we choose the full-scale LCD period for both 2020 and 2022.

Accordingly, we have modified the following content to clarify:

"The decrease in $NO_2$ concentrations (49%) during the 2022 LCD was larger than the $NO_2$ decrease during the full-scale LCD phase in 2020 (23 January–12 February), when the average $NO_2$ concentration decreased by 30.5% from the corresponding period in 2019 (Fig. S3b)."

Reference:
Liu, Y., Wang, T., Stavrakou, T., Elguindi, N., Doumbia, T., Granier, C., Bouarar, I., Gaubert, B., and Brasseur, G. P.: Diverse response of surface ozone to COVID-19 lockdown in China, Science of the Total Environment, 789, https://doi.org/10.1016/j.scitotenv.2021.147739, 2021.
Ding, D., Cheng, M., del Pozo Cruz, B., Lin, T., Sun, S., Zhang, L., Yang, Q., Ma, Z., Wang, J., Jia, Y., and Shi, Y.: How COVID-19 lockdown and reopening affected daily steps: evidence based on 164,630 person-days of prospectively collected data from Shanghai, China, Int J Behav Nutr Phys Act, 18, 40, https://doi.org/10.1186/s12966-021-01106-x, 2021.

**L209 (Fig.9): in this picture, your x-axis is $HCHO/NO_2$ ratio, and y-axis is MDA8 $O_3$ (ug/m$^3$), but OMI $HCHO/NO_2$ and high-$O_3$ Probability (%) in your reference paper, respectively (Wang et al., 2021). Can you explain this difference? Weather the change of axis will affect $O_3$ formation regimes?**
Response: Thanks for your comment. The essence of using MDA8 $O_3$ (ug/m$^3$) or high-$O_3$ Probability (%) as the y-axis are similar, both to examine the $HCHO/NO_2$ ratio threshold above which the trend of the surface $O_3$ concentration starts to decrease with increasing $HCHO/NO_2$ ratio.

In the papers we cited, Jin et al. (2020) used the high-$O_3$ Probability (%) as the y-axis, which refers to the possibility of observations with $O_3$ higher than 70 ppbv and defined the range over the top 10% of the fitted curve as the transition regime. While Wang et al. (2021) defined the transition regime as the $\pm$ 30 % range from the median of monthly $O_3$ (13:00-14:00 LT) exceeding 160 ug/m$^3$ in the top 10% dataset. Another recent study utilized the daily $HCHO/NO_2$ ratio coupled with daily surface MDA8 $O_3$ data to indicate the $O_3$ formation regime according to the relationship between

HCHO/$NO_2$ ratio and MDA8 $O_3$ (Schroeder et al., 2022). So, there is not a commonly agreed definition and calculation method of the HCHO/$NO_2$ threshold of the transition regime. But the principles of these three studies are the same, all to examine the $O_3$ formation regime based on the relationship between the HCHO/$NO_2$ ratio and the surface $O_3$ concentration.

In our study, we chose the MDA8 $O_3$ as the y-axis to select the high $O_3$ values from the dataset and examine its relationship with the HCHO/$NO_2$ ratio. The thresholds that we obtained in this way are consistent with those of the above studies. Although there may be difference among the different methods and selected ranges, the essence does not change, and we believe using MDA8 $O_3$ (ug/m$^3$) as the y-axis is a straightforward and scientifically justified way to determine of the $O_3$ formation regime.

In our manuscript, we added another citation which also used MDA8 $O_3$ as the indicator as follows:

"To determine the regime transition threshold, we adopted an observation-based method similar to those used in previous studies (Jin et al., 2020; Wang et al., 2021; Schroeder et al., 2022)."

Reference:

Jin, X., Fiore, A., Boersma, K. F., Smedt, I. D., and Valin, L.: Inferring Changes in Summertime Surface Ozone–NO x –VOC Chemistry over U.S. Urban Areas from Two Decades of Satellite and Ground-Based Observations, Environ. Sci. Technol., 54, 6518–6529, https://doi.org/10.1021/acs.est.9b07785, 2020.

Wang, W., van der A, R., Ding, J., van Weele, M., and Cheng, T.: Spatial and temporal changes of the ozone sensitivity in China based on satellite and ground-based observations, Atmospheric Chemistry and Physics, 21, 7253–7269, https://doi.org/10.5194/acp-21-7253-2021, 2021.

Schroeder, J. R., Cai, C., Xu, J., Ridley, D., Lu, J., Bui, N., Yan, F., and Avise, J.: Changing ozone sensitivity in the South Coast Air Basin during the COVID-19 period, Atmospheric Chemistry and Physics, 22, 12985–13000, https://doi.org/10.5194/acp-22-12985-2022, 2022.